# TacTape: Real-time High-accuracy Tactile Fiducial System with Structured 3D Texture for Vision-based Tactile Sensors

Meng Wang[1*], Wanlin Li[1*], Qiuxuan Chen[1,2], Yuzhe Huang[1,3], Hang Li[1],
Kaspar Althoefer[4], Ziyuan Jiao[1], Yao Su[1†], and Hangxin Liu[1†]

*Abstract*— **Vision-based tactile sensors enable high-resolution tactile perception by capturing image-based contact data. However, their utility in tactile localization is limited by their inherently small and local sensing area, as well as their dependence on distinct object surface features. We propose TacTape, a novel tactile fiducial system that enables accurate and efficient tactile localization by attaching textured tape to object surfaces. A lightweight algorithm allows real-time estimation of contact position and orientation from partially observed structured 3D textures. Experiments demonstrate that TacTape achieves sub-millimeter positional and sub-degree angular localization accuracy, and operates significantly faster than classic tactile mapping methods.**

## I. INTRODUCTION

Vision-based tactile sensors (VBTSs) [1–12] provide pixel-level contact imaging, enabling high-resolution tactile perception that supports diverse robotic manipulation tasks, including shear/slip force regulation [7, 13], texture classification [14, 15], object manipulation [16–18], and the handling of fragile and deformable objects [5, 19, 20]. These advances are largely supported by the ability to localize contact using high-resolution tactile information. While traditional tactile sensing relies on distinct surface features and struggles with smooth or repetitive textures, tactile fiducial systems—analogous to visual fiducials—offer a promising alternative by embedding structured patterns for direct localization and identification. However, unlike visual systems with global tag capture, VBTSs only perceive local contacts, necessitating fiducials that can be robustly decoded from partial observations. This challenge underscores the need for parameterized, fully automated pipelines for designing and fabricating 3D structured tactile tags.

In this paper, we propose TacTape, a tactile fiducial system that enables accurate and real-time tactile localization on objects for VBTSs. TacTape features a flexible tape that can be easily customized to apply structured textures to objects. It is fabricated using standard SMT-FPC (Surface Mount Technology Flexible Printed Circuit) through a parameterized and automated process, ensuring high reproducibility

* Meng Wang and Wanlin Li contributed equally to this work.
† Corresponding authors.
[1] State Key Laboratory of General Artificial Intelligence, Beijing Institute for General Artificial Intelligence (BIGAI). Emails: {wangmeng, liwanlin, chenqiuxuan, huangyuzhe, lihang, jiaoziyuan, suyao, liuhx}@bigai.ai.
[2] School of Artificial Intelligence, Beijing University of Posts and Telecommunications.
[3] Department of Automation, Beihang University.
[4] Centre for Advanced Robotics @ Queen Mary (ARQ), Queen Mary University of London.

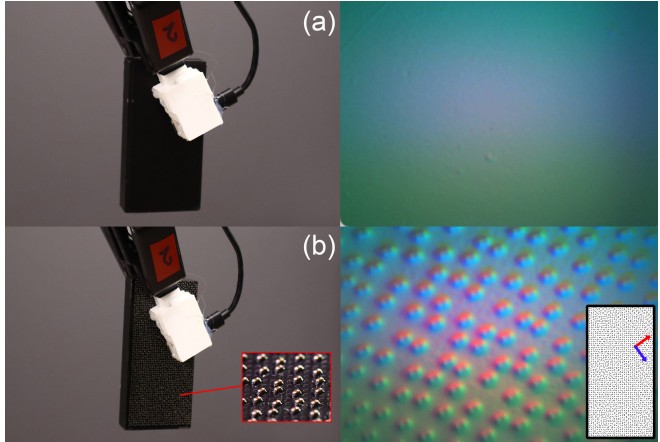

Fig. 1: **Tactile localization with TacTape.** (a) Vision-based tactile sensors such as GelSight Mini [21] fail to localize flat objects lacking surface features. (b) TacTape overlays structured 3D patterns to enable localization from partial contact. Left inset: magnified surface pattern. Right inset: decoded contact position and orientation.

and consistency while supporting scalable manufacturing. By integrating a symbolic encoding and decoding scheme, a geometric method for efficient pattern recognition, and pixel resolution transformations, the system achieves sub-millimeter positional and sub-degree angular localization accuracy. The fabrication and localization pipeline is illustrated in Fig. 2. TacTape's real-time, high-accuracy performance and ease of deployment make it well-suited for laboratory and potential industrial setups and provide a reliable metric for evaluating and adjusting gripping actions (shown in Fig. 4) with VBTSs.

## II. CHARACTERISTICS OF TACTAPE

TacTape features a flexible base with a solid texture and is fabricated using FPC-SMT technology. Resistors are mounted to create the structured 3D texture. Given the physical dimension of the **grid space** $W$, a symbolic matrix can be generated via *py-microdots* [22], ensuring positional decoding by utilizing De Bruijn sequences. Given the **physical displacement** $\Delta$, layered CAD files for PCB fabrication can be automatically rendered as the symbolic matrix for the placement of all components. Given the footprint, the components are rotated by $45°$ to avoid pattern overlap.

A 4-axis XYZR linear stage trimming platform is used to evaluate the TacTape's performance with the use of a GelSight Mini mounted on the Z axis slider. During the evaluation, a minimum of $2\ N$ force is necessary

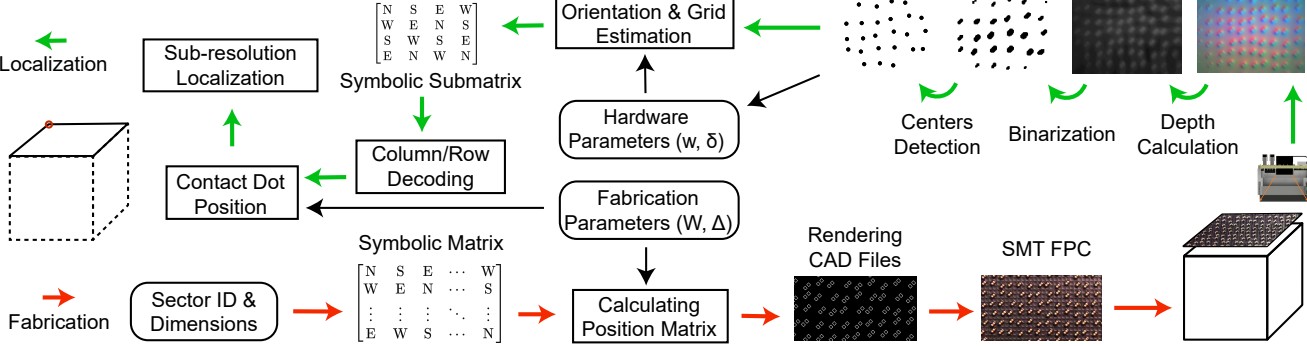

Fig. 2: **Fabrication and localization pipeline of TacTape**. Given a region-specific ID, dimensions, and sensor-specific fabrication parameters, the texture engine automatically generates an encoded symbolic matrix and renders the corresponding CAD files for SMT FPC. Mounted solid components form a structured texture that can be captured by vision-based tactile sensors (VBTSs) and transformed into structured 2D patterns through depth binarization. By estimating the orientation and restoring the grid, a symbolic submatrix can be extracted from the partially observed pattern. After decoding the column and row indices of the submatrix in the original symbolic matrix, the sensor's position within the specified area can be accurately estimated using a subsequent sub-resolution transform.

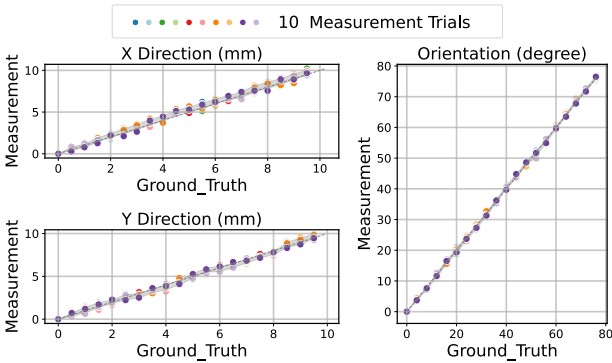

Fig. 3: **Linearity and repeatability of the TacTape**. Ten trials of accuracy and repeatability test were conducted for translations in x and y directions (from 0 to $10mm$ with a step of $0.5mm$) and rotation r in z-direction (from 0 to $80°$ with a step of $4°$).

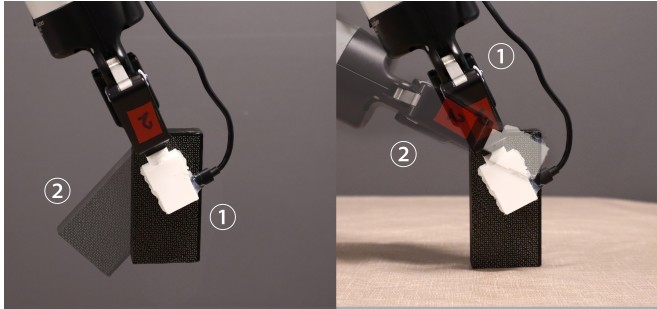

Fig. 4: **Robust Human-Robot Interaction**. Given different initial grasping states ① at the center or ② at the corner during the handover, the robot is able to adapt and place the object securely in the same position.

to capture sufficient patterns for successful decoding. The tape is tested to support a minimum curvature radius of $15.7\ mm$, corresponding to a curvature of $63.69\ m^{-1}$. Fig. 3 shows the measurement trails, where the root mean square error (RMSE) is $0.23\ mm$, $0.19\ mm$, and $0.3°$; the R-square values of linear regression are 0.99, 0.99, and 0.99; the repeatability, calculated as the standard deviation, are $0.26\ mm$, $0.23\ mm$, $0.5°$ in XYR on average. The algorithm achieves a decoding success rate of over 99% given sufficient input. Localization algorithm only takes less than $3\ ms$ for each frame, excluding time needed for depth reconstruction ($30\ ms$ in our implementation).

## III. DISCUSSION

**Deployment limitation**. For non-developable surfaces such as spheres, tapes may not adhere perfectly. For such complex geometries, we suggest integrating the patterns during the fabrication process, such as 3D printing or milling with CNC. Nevertheless, planar and cylindrical surfaces account for the majority of interactive areas on everyday objects, making TacTape highly applicable in practical scenarios.

**Fiducial application**. By decoupling localization from object geometry or learning-based priors, TacTape enables plug-and-play manipulation across diverse tasks, tools, and surfaces. This opens up new possibilities for general-purpose tactile augmentation in unstructured environments. By integrating an additional visual fiducial marker, TacTape establishes a shared reference for both vision and touch, enabling accurate, continuous, and cross-modal interaction in real-world robotic systems. Furthermore, when combined with force estimation from depth using conventional methods, the system supports closed-loop control of both force and position during manipulation.

## IV. CONCLUSION

This paper introduces TacTape, a novel tactile fiducial system featuring a flexible, textured tape that enables easy modification of object surfaces to enhance interaction with VBTSs. By employing encoded and structured 3D textures, contact position and orientation can be directly calculated even for objects with smooth surfaces. Combined with the proposed algorithm, TacTape outperforms existing tactile localization methods in both accuracy and efficiency. TacTape is fabricated using standard FPC-SMT technology through a parameterized and automated pipeline, ensuring high reproducibility and accessibility across a wide range of scenarios.

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
