# OpenReview forum: "TacTape: Real-time High-accuracy Tactile Fiducial System with Structured 3D Texture for Vision-based Tactile Sensors"
_IEEE.org/IROS/2025/Workshop/Tactile_Sensing — IROS 2025 Workshop Tactile Sensing Poster_

### Official Review · Reviewer_hfgc · 2025-09-15
**Good ideas but needs more work**

**Rating:** 6
**Confidence:** 4

**Review:**

This paper promotes the novel idea of using tape for vision based sensing. This idea has great merit and could make setting up optical sensors easier for a range of shapes and sizes. The paper is well written and has a good discussion about limitations of the sensor itself. The paper itself could benefit from several revisions.

1. Somewhere the tactile tape should be shown in its tape form of how it is applied. Figure 2 would probably be the place to do this. At the moment images of the tape are flat and not promoting the novelty of this sensor.
2. Figure 3 should be clearer on what measurement is. What unit is measurement in, RMSE? State this in the figure caption if it does not fit on the axis itself.
3. To take this paper further the author should consider testing this sensor across a range of common benchmark tasks such as texture classification (https://ieeexplore.ieee.org/abstract/document/8250220), and with resolution investigations (https://www.mdpi.com/1424-8220/25/16/4971) or edge following (https://ieeexplore.ieee.org/document/8641397)
4. The resistors on the tape are unclear on why they are there. Is this purely to give a 3D shape but are not used as resistors? If so surely it would be simpler and cheaper to make moulds for silicone tape that had these 3D features (similar to the TacTip).

---

### Official Review · Reviewer_HKtF · 2025-09-23
**Review for submission #15**

**Rating:** 6
**Confidence:** 3

**Review:**

The paper proposes TacTape, a tactile fiducial system to facilitate tactile localization by attaching textured tape to object surfaces. The tape is fabricated by SMT-FPC (Surface Mount Technology Flexible Printed Circuit).

Strengths
* It is a cool idea to attach a tape to object surfaces to facilitate tactile localization, especially objects with smooth and textureless surfaces.
* The fabrication process of the TacTape is designed to be automatic and efficient.
* Figures in the paper are very illustrative.

Questions/Weakness
* The repeatability of the tape applied to different objects is unclear. How does it work with objects made of various materials? How does it work with objects with irregular shapes and sharp angles?
* As claimed in Sec III, the TacTape doesn't work well with non-developable surfaces like a sphere. This hugely constrains the application of this tape.
* One main benefit of VBTS is that the sensor can capture the fine details of the object and detect the minimal amount of force or torque via surface deformation when interacting with the object. If we now cover all objects with TacTape, the original information from interacting with the object will be missing.

---

### Official Review · Reviewer_yoH6 · 2025-09-24
**Good idea but needs more work**

**Rating:** 6
**Confidence:** 3

**Review:**

This paper presents a tactile fiducial system fabricated using standard SMT-FPC (Surface-Mount Technology Flexible Printed Circuit) processes. Through a parameterized and automated pipeline, the system enables real-time tactile localization for VBTSs.

The reviewer would like to offer the following suggestions for improvement:
(1) It would be helpful to include a more detailed demonstration of the fabrication process of the proposed sensor, which could make it easier for readers to follow and understand.
(2) More convincing characterization experiments would strengthen the validation of the design’s rationality.
(3) Additional demonstration experiments could further highlight the effectiveness and practical value of the proposed design.